# ACHIEVING EXPLAINABILITY IN A VISUAL HARD ATTENTION MODEL THROUGH CONTENT PREDICTION

## ABSTRACT

A visual hard attention model actively selects and observes a sequence of subregions in an image to make a prediction. Unlike in the deep convolution network, in hard attention it is explainable which regions of the image contributed to the prediction. However, the attention policy used by the model to select these regions is not explainable. The majority of hard attention models determine the attention-worthy regions by first analyzing a complete image. However, it may be the case that the entire image is not available in the beginning but instead sensed gradually through a series of partial observations. In this paper, we design an efficient hard attention model for classifying partially observable scenes. The attention policy used by our model is explainable and non-parametric. The model estimates expected information gain (EIG) obtained from attending various regions by predicting their content ahead of time. It compares EIG using Bayesian Optimal Experiment Design and attends to the region with maximum EIG. We train our model with a differentiable objective, optimized using gradient descent, and test it on several datasets. The performance of our model is comparable to or better than the baseline models.

## 1 INTRODUCTION

Though deep convolution networks achieve state of the art performance on the image classification task, it is difficult to explain which input regions affected the output. A technique called visual hard attention provides this explanation by design. The hard attention model sequentially attends small but informative subregions of the input called glimpses to make predictions. While the attention mechanism explains the task-specific decisions, the attention policies learned by the model remain unexplainable. For example, one cannot explain the attention policy of a caption generation model that correctly predicts the word 'frisbee' while looking at a region far from an actual frisbee (Xu et al. (2015)).

The majority of hard attention models first analyze a complete image to locate the task-relevant subregions and then attend to these locations to make predictions (Ba et al. (2014); Elsayed et al. (2019)). However, in practice, we often do not have access to the entire scene, and we gradually attend to the important subregions to collect task-specific information. At each step in the process, we decide the next attention-worthy location based on the partial observations collected so far. The explainable attention policies are more desirable under such partial observability.

Pioneering work by Mnih et al. (2014) presents a model that functions under partial observability but their attention policies are not explainable. They train their model with the REINFORCE algorithm (Williams (1992)), which is challenging to optimize. Moreover, the model's performance is affected adversely if the parameterization of the attention policy is not optimal. For example, an object classification model with unimodal Gaussian policy learns to attend the background region in the middle of the two objects (Sermanet et al. (2014)).

This paper develops a hard-attention model with an explainable attention policy for classifying images through a series of partial observations. We formulate the problem of hard attention as a Bayesian Optimal Experiment Design (BOED). A recurrent model finds an optimal location that gains maximum expected information about the class label and attends to this location. To estimate expected information gain (EIG) under partial observability, the model predicts content of the un-

seen regions based on the regions observed so far. Using the knowledge gained by attending various locations in an image, the model predicts the class label.

To the best of our knowledge, ours is the first hard attention model that is entirely explainable under partial observability. Our main contributions are as follows. First, our attention policies are explainable by design. One can explain that the model attends a specific location because it expects the corresponding glimpse to maximize the expected information gain. Second, the model does not rely on the complete image to predict the attention locations and provides good performance under partial observability. Third, the training objective is differentiable and can be optimized using standard gradient backpropagation. We train the model using discriminative and generative objectives to predict the label and the image content, respectively. Fourth, our attention policy is non-parametric and can be implicitly multi-modal.

## 2 RELATED WORKS

A hard attention model prioritizes task-relevant regions to extract meaningful features from an input. Early attempts to model attention employed image saliency as a priority map. High priority regions were selected using methods such as winner-take-all (Koch & Ullman (1987); Itti et al. (1998); Itti & Koch (2000)), searching by throwing out all features but the one with minimal activity (Ahmad (1992)), and dynamic routing of information (Olshausen et al. (1993)).

Few used graphical models to model visual attention. Rimey & Brown (1991) used augmented hidden Markov models to model attention policy. Larochelle & Hinton (2010) used a Restricted Boltzmann Machine (RBM) with third-order connections between attention location, glimpse, and the representation of a scene. Motivated by this, Zheng et al. (2015) proposed an autoregressive model to compute exact gradients, unlike in an RBM. Tang et al. (2014) used an RBM as a generative model and searched for informative locations using the Hamiltonian Monte Carlo algorithm.

Many used reinforcement learning to train attention models. Paletta et al. (2005) used Q-learning with the reward that measures the objectness of the attended region. Denil et al. (2012) estimated rewards using particle filters and employed a policy based on the Gaussian Process and the upper confidence bound. Butko & Movellan (2008) modeled attention as a partially observable Markov decision process and used a policy gradient algorithm for learning. Later, Butko & Movellan (2009) extended this approach to multiple objects.

Recently, the machine learning community use the REINFORCE policy gradient algorithm to train hard attention models (Mnih et al. (2014); Ba et al. (2014); Xu et al. (2015); Elsayed et al. (2019)). Among these, only Elsayed et al. (2019) claims explainability by design. Other works use EM-style learning procedure (Ranzato (2014)), wake-sleep algorithm (Ba et al. (2015)), a voting based region selection (Alexe et al. (2012)), and differentiable models (Gregor et al. (2015); Jaderberg et al. (2015); Eslami et al. (2016)).

Among the recent models, Ba et al. (2014); Ranzato (2014); Ba et al. (2015) look at the low-resolution gist of an input at the beginning, and Xu et al. (2015); Elsayed et al. (2019); Gregor et al. (2015); Jaderberg et al. (2015); Eslami et al. (2016) consume the whole image to predict the locations to attend. In contrast, our model does not look at the entire image at low resolution or otherwise. Moreover, our attention policies are explainable. We can apply our model in a wide range of scenarios where explainable predictions are desirable for the partially observable images.

## 3 MODEL

In this paper, we consider a recurrent attention model that sequentially captures glimpses from an image $x$ and predicts a label $y$. The model runs for time $t = 0$ to $T - 1$. It uses a recurrent net to maintain a hidden state $h_{t-1}$ that summarizes glimpses observed until time $t - 1$. At time $t$, it predicts coordinates $l_t$ based on the hidden state $h_{t-1}$ and captures a square glimpse $g_t$ centered at $l_t$ in an image $x$, i.e. $g_t = g(x, l_t)$. It uses $g_t$ and $l_t$ to update the hidden state to $h_t$ and predicts the label $y$ based on the updated state $h_t$.

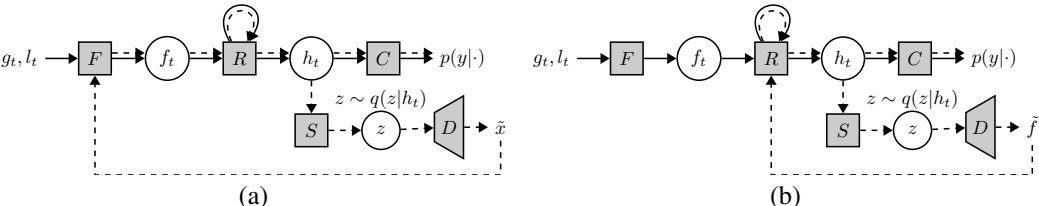

Figure 1: A recurrent attention model sequentially observes glimpses from an image and predicts a class label. At time $t$, the model actively observes a glimpse $g_t$ and its coordinates $l_t$. Given $g_t$ and $l_t$, the feed-forward module $F$ extracts features $f_t$, and the recurrent module $R$ updates a hidden state to $h_t$. Using an updated hidden state $h_t$, the linear classifier $C$ predicts the class distribution $p(y|h_t)$. At time $t + 1$, the model assesses various candidate locations $l$ before attending an optimal one. It predicts $p(y|g, l, h_t)$ ahead of time and selects the candidate $l$ that maximizes $KL[p(y|g, l, h_t)||p(y|h_t)]$. (a) The model predicts the content of $g$ using a Partial VAE to compute $p(y|g, l, h_t)$ without attending to the glimpse $g$. The normalizing flow-based encoder $S$ predicts the approximate posterior $q(z|h_t)$, and the decoder $D$ predicts an image $\tilde{x}$ from a sample $z \sim q(z|h_t)$. The model uses glimpses from the predicted image $\tilde{x}$ to evaluate $p(y|g, l, h_t)$. Dashed arrows show a path to compute the lookahead class distribution $p(y|g, l, h_t)$. (b) Alternatively, the model predicts $\tilde{f}$, a feature representation of the image $x$. In $\tilde{f}$, the features at location $l$ are the features of the glimpse at location $l$. The model uses $\tilde{f}(l)$ to predict the lookahead class distribution $p(y|g, l, h_t) \approx p(y|\tilde{f}(l), h_t)$. Predicting glimpse content in feature space shortens the lookahead path.

### 3.1 ARCHITECTURE

As shown in Figure 1(a), our model comprises the following three building blocks. A recurrent feature aggregator ($F$ and $R$) maintains a hidden state $h_t$. A classifier ($C$) predicts the class probabilities $p(y|h_t)$. A normalizing flow-based variational autoencoder ($S$ and $D$) predicts a complete image given the hidden state $h_t$; a flow-based encoder $S$ predicts the posterior of a latent variable $z$ from $h_t$, and a decoder $D$ predicts a complete image from $z$. The BOED, as discussed in section 3.2, uses the predicted image to find an optimal location to attend at the next time-step. To distinguish the predicted image from the actual image, let us call the former $\tilde{x}$. Henceforth, we crown any quantity derived from the predicted image $\tilde{x}$ with a ($\tilde{\ }$). Next, we provide details about the three building blocks of the model, followed by a discussion of the BOED in the context of hard attention.

#### 3.1.1 A RECURRENT FEATURE AGGREGATOR

Given a glimpse $g_t$ and its location $l_t$, a feed-forward module extracts features $f_t = F(g_t, l_t)$, and a recurrent network updates a hidden state to $h_t = R(h_{t-1}, f_t)$. Following Mnih et al. (2014), we define $F(g_t, l_t) = BN(LeakyReLU(F_g(g_t) + F_l(l_t)))$ where $F_g$ and $F_l$ are deep networks, and $R(h_{t-1}, f_t) = LN(LeakyReLU(Linear(h_{t-1}) + Linear(f_t)))$. Here, $BN$ is a BatchNorm layer (Ioffe & Szegedy (2015)) and $LN$ is a LayerNorm layer (Ba et al. (2016)).

#### 3.1.2 A CLASSIFIER

At each time-step $t$, a linear classifier predicts the distribution $p(y|h_t) = C(h_t)$ from a hidden state $h_t$. As the goal of the model is to predict a label $y$ for an image $x$, we learn a distribution $p(y|h_t)$ by minimizing $KL[p(y|x)||p(y|h_t)]$. Optimization of this KL divergence is equivalent to the minimization of the following cross-entropy loss.

$$\mathcal{L}_{CE}(t) = -p(y|x)\log(p(y|h_t)) \tag{1}$$

#### 3.1.3 A PARTIAL VARIATIONAL AUTOENCODER

We adapt a variational autoencoder (VAE) to predict the complete image $x$ from the hidden state $h_t$. A VAE learns a joint distribution between the image $x$ and the latent variable $z$ given $h_t$, $p(x, z|h_t) = p(x|z)p(z|h_t)$. An amortized encoder infers the posterior $q(z|x, h_t)$, which is an approximation of the true posterior $p(z|x, h_t)$, and a decoder infers the likelihood $p(x|z)$. The training of VAE requires optimizing the Evidence Lower Bound (ELBO), which involves calculating $KL[q(z|x, h_t)||p(z|h_t)]$ (Kingma & Welling (2013)). As the hard attention model does not observe

the complete image $x$, it cannot estimate $q(z|x, h_t)$. Hence, we cannot incorporate the standard VAE directly into a hard attention framework.

At the time $t$, we separate an image $x$ into two parts, $o_t$ — the set of regions observed up to $t$, and $u_t$ — the set of regions as yet unobserved. Ma et al. (2018) observed that in a VAE, $o_t$ and $u_t$ are conditionally independent given $z$, i.e. $p(x|z) = p(u_t|z)p(o_t|z)$. They predict $u_t$ independently from the sample $z \sim q(z|o_t)$, while learning the approximate posterior $q(z|o_t)$ by optimizing the ELBO on $\log(p(o_t))$. They refer to the resultant VAE as a Partial VAE. Recall that the hidden state $h_t$ calculated by our attention model is a summary of the glimpses observed up to $t$, which is equivalent to $o_t$, the set of observed regions. Hence, we can write $q(z|o_t)$ as $q(z|h_t)$ in the ELBO of the Partial VAE.

$$\mathcal{L}_{\text{PVAE}}(t) = \mathbb{E}_{q(z|o_t)} \log(p(o_t|z)) - KL[q(z|o_t)||p(z)]$$
$$= \mathbb{E}_{q(z|h_t)} \log(p(o_t|z)) - KL[q(z|h_t)||p(z)] \quad (2)$$

In a Partial VAE, $p(x, z|h_t) = p(u_t|z)p(o_t|z)p(z|h_t)$. We implement a decoder $D$ that predicts the complete image given the sample $z \sim q(z|h_t)$. Let $m_t$ be a binary mask with value **1** for the pixels observed by the model up to $t$ and **0** otherwise; hence, $o_t = m_t \odot x$, where $\odot$ is an element-wise multiplication. We write the log-likelihood in equation 2 using the mask $m_t$ as follows.

$$\log(p(o_t|z)) = -0.5 \sum |m_t \odot D(z) - m_t \odot x|^2 = -0.5 \sum m_t \odot |D(z) - x|^2 \quad (3)$$

In equation 2, the prior $p(z)$ is a Gaussian distribution with zero mean and unit variance. To obtain expressive posterior $q(z|h_t)$, we use normalizing flows(Kingma et al. (2016)). As an explicit inversion of the flows is not required, we use auto-regressive Neural Spline Flows (NSF) (Durkan et al. (2019)) and efficiently implement them using a single feed-forward network with masked weights as in De Cao et al. (2019). Between the two flow layers, we flip the input (Dinh et al. (2016)) and normalize it using ActNorm (Kingma & Dhariwal (2018)). In Figure 1(a), the flow-based encoder $S$ infers the posterior $q(z|h_t) = S(h_t)$. As mentioned earlier, we refer to the prediction from the Partial VAE as $\tilde{x}$. The BOED uses $\tilde{x}$ to find an optimal location to attend. [1]

## 3.2 BAYESIAN OPTIMAL EXPERIMENT DESIGN (BOED)

The BOED evaluates the optimality of a set of experiments by measuring the information gain in the interest parameter due to the experimental outcome. (Chaloner & Verdinelli (1995)). In the context of hard attention, an experiment is to attend a location $l$ and observe a corresponding glimpse $g = g(x, l)$. An experiment of attending a location $l$ is optimal if it gains maximum information about the class label $y$. We can evaluate the optimality of attending a specific location by measuring several metrics such as feature variance (Huang et al. (2018)), uncertainty in the prediction (Melville et al. (2004)), expected Shannon information (Lindley (1956)). For a sequential model, information gain $KL[p(y|g, l, h_{t-1})||p(y|h_{t-1})]$ is an ideal metric. It measures the change in the entropy of the class distribution from one time-step to the next due to observation of a glimpse $g$ at location $l$ (Bernardo (1979); Ma et al. (2018)).

The model has to find an optimal location to attend at time $t$ before observing the corresponding glimpse. Hence, we consider an expectation of information gain over the generating distribution of $g$. An expected information gain (EIG) is also a measure of Bayesian surprise (Itti & Baldi (2006); Schwartenbeck et al. (2013)).

$$EIG(l) = \mathbb{E}_{p(g|l, h_{t-1})} KL[p(y|g, l, h_{t-1})||p(y|h_{t-1})] \quad (4)$$

Inspired by Harvey et al. (2019), we define the distribution $p(g|l, h_{t-1})$ as follows.

$$p(g|l, h_{t-1}) = \mathbb{E}_{q(z|h_{t-1})} p(g|z, l) = \mathbb{E}_{q(z|h_{t-1})} \delta(g(D(z), l)) \quad (5)$$

Here, $\delta(\cdot)$ is a delta distribution. As discussed in the section 3.1.3, the flow-based encoder $S$ predicts the posterior $q(z|h_{t-1})$ and the decoder $D$ predicts the complete image $\tilde{x}$. $g(\cdot)$ extracts a glimpse located at $l$ in the predicted image $\tilde{x} = D(z)$. Combining equation 4 and equation 5 yields,

$$EIG(l) = \mathbb{E}_{q(z|h_{t-1})} KL[p(y|g(D(z), l), l, h_{t-1})||p(y|h_{t-1})] \quad (6)$$

---

[1]We also tried an alternate method to predict the image $x$ given $h_t$ as discussed in section C.

To find an optimal location to attend at time $t$, the model compares various candidates for $l_t$. It predicts $EIG(l)$ for each candidate $l$ and selects an optimal candidate as $l_t$, i.e. $l_t = \arg\max_l EIG(l)$.

When the model is considering a candidate $l$, it predicts $\tilde{f} = F(g(\tilde{x}, l), l)$, $\tilde{h}_t = R(h_{t-1}, \tilde{f})$ and $p(y|\tilde{h}_t) = C(\tilde{h}_t)$. It uses the distribution $p(y|\tilde{h}_t) = p(y|g(D(z), l), l, h_{t-1})$ to predict $EIG$ in equation 6. We refer to $p(y|\tilde{h}_t)$ as the *lookahead* class distribution computed by anticipating the content at the location $l$ ahead of time. In Figure 1(a), the dashed arrows show a lookahead step.

### 3.2.1 EFFICIENT COMPUTATION OF EIG

*Convolutional implementation:* To compute $EIG$ for all locations simultaneously, we implement all modules of our model with convolution layers. The model computes $EIG$ for all locations as a single activation map in a single forward pass. An optimal location is equal to the coordinates of the pixel with maximum value in the $EIG$ map.

*Feature generation:* During the lookahead step, the feature extractor $F$ computes a feature map $\tilde{f}$ from the predicted image $\tilde{x}$. However, we can use the decoder $D$ to directly predict the feature map $\tilde{f}$. Predicting $\tilde{f}$ instead of $\tilde{x}$ achieves two goals. First, the time to compute $EIG$ is reduced as the model does not require a feature extractor $F$ during the lookahead step (see Figure 1(b)). Second, the Partial VAE does not have to produce unnecessary details that may later be thrown away by the feature extractor $F$, such as the exact color of a pixel.

In most cases, the number of elements in the feature map $\tilde{f}$ is greater than the elements in $\tilde{x}$. The model requires more memory resources and parameters to predict $\tilde{f}$. So instead, our model predicts a subsampled feature map that contains features for every $n^{th}$ glimpse. Consequentially, the model computes an $EIG$ map at low resolution and finds optimal glimpses from a subset of glimpses separated with stride equal to $n$.

When the decoder predicts a feature map $\tilde{f}$ instead of an image $\tilde{x}$, the feature likelihood $p(f_{1:t}|z)$ replaces the likelihood $p(o_t|z)$ in equation 2. $f_{1:t}$ are the features of the glimpses observed by the model until time $t$. Similar to equation 3, the corresponding log-likelihood is computed as follows.

$$\log(p(f_{1:t}|z)) = -0.5 \sum m_t \odot |D(z) - f|^2 \tag{7}$$

Here, $f$ is a map containing the features of all glimpses. As $\tilde{f}$ is a feature map of strided glimpses, we subsample the mask $m_t$ and the feature map $f$ in the above log-likelihood.

## 4 TRAINING AND TESTING

The training objective is $\mathcal{L} = \sum_{t=0}^{T-1} \mathcal{L}_{\text{PVAE}}(t) + \lambda \mathcal{L}_{CE}(t)$, where $\lambda$ is a hyperparameter. The training procedure is given in section B. We use only one sample $z \sim q(z|h_t)$ to estimate the $EIG$ map during the training. We found this to work well in practice. We do not compute gradients and do not update the statistics of normalization layers during the *lookahead* step. When the decoder predicts the feature map instead of the image, we pre-train modules $F$, $R$, and $C$ for few epochs with random glimpses. This provides a reasonable target $f$ for the log-likelihood in equation 7. We naturally achieve exploration early in the training when the model produces noisy output and computes noisy $EIG$ maps. As the training progress and the predictions become accurate, the exploitation begins. The testing procedure is shown in Algorithm 1. Unlike in the training phase, we use $P$ samples of $z$ to estimate the $EIG$ accurately during the prediction phase.

---
**Algorithm 1** Test procedure for the model shown in Figure 1(b)
---
1:  Randomly sample $l_0$; Capture $g_0$ at $l_0$, compute $f_0$, $h_0$ and $p(y|h_0)$
2:  **for** $t \in \{1, \ldots, T-1\}$ **do**                    ▷ T is the time budget
3:      Sample $z_i \sim q(z|h_{t-1})$ and predict $\tilde{f}^i$; $i \in \{0, \ldots, P-1\}$     ▷ P is the sample budget
4:      Compute $\tilde{h}_t^i$, $p(y|\tilde{h}_t^i)$ and $EIG = \frac{1}{P}\sum_i KL[p(y|\tilde{h}_t^i)||p(y|h_{t-1})]$     ▷ equation 6
5:      $l_t = \arg\max\{EIG\}$
6:      Capture $g_t$ at $l_t$; Compute $f_t$, $h_t$ and $p(y|h_t)$
7:  **end for**
---

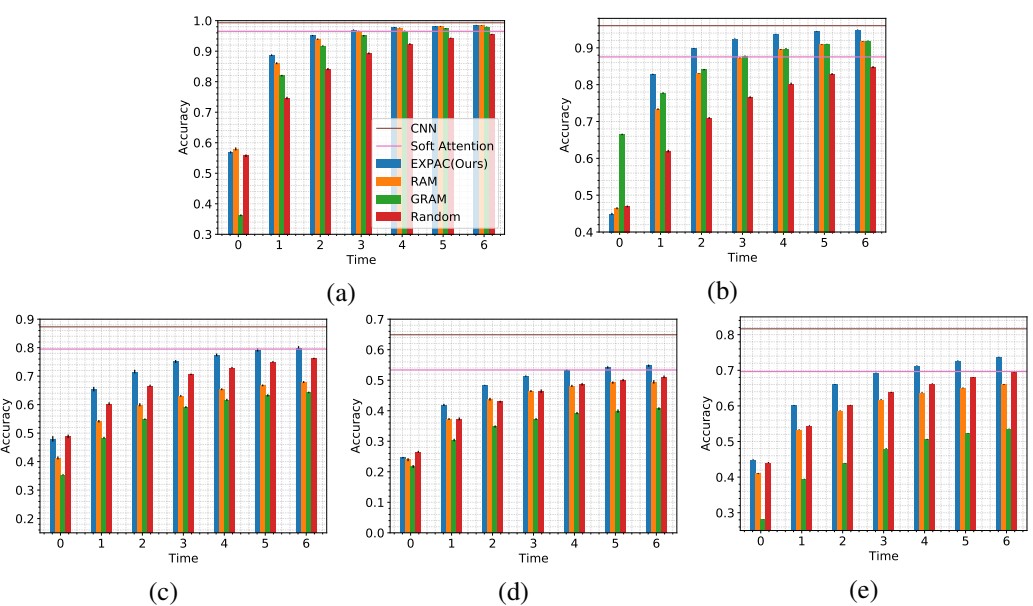

Figure 2: *The classification accuracy as a function of time.* A CNN and a soft attention model observe the entire image to predict the class label. EXPAC, RAM(Mnih et al. (2014)), and GRAM sequentially attends glimpses in the image and predict the class label. EXPAC and RAM never observe a complete image and attend the first glimpse at a random location. GRAM observes a gist of an input at the beginning and predicts glimpse locations from $t = 0$. (a) MNIST (b) SVHN (c) CIFAR-10 (d) CIFAR-100 (e) CINIC-10.

## 5 EXPERIMENTS

We refer to our model as 'EXPAC' which stands for *EXP*lainable *A*ttention with *C*ontent prediction. We evaluate EXPAC on MNIST (LeCun et al. (1998)), SVHN (Netzer et al. (2011)), CIFAR-10, CIFAR-100 (Krizhevsky et al. (2009)), and CINIC-10 (Darlow et al. (2018)) datasets. The MNIST and the SVHN datasets consist of gray-scale and color images of the 0-9 digits, respectively. The CINIC-10, CIFAR-10 and CIFAR-100 datasets are composed of color images of real-world objects in the natural setting, categorized into 10, 10 and 100 classes, respectively.

For the MNIST dataset, we use the model shown in Figure 1(a). This model predicts image $\tilde{x}$ and $EIG$ at full resolution and finds optimal glimpse from a set of all overlapping glimpses. We use the model shown in Figure 1(b) for the remaining datasets. This model predicts $\tilde{f}$ and $EIG$ for a set of glimpses separated with stride equal to $n$. We do not allow this model to revisit glimpses attended in the past. All models run for $T = 7$ time-steps and sense glimpses of size $8 \times 8$ with stride $n = 4$. The hyperparameter $\lambda$ in the training objective is 500, and the sample budget $P$ is 20 for all experiments. We did not observe any performance gain with $P > 20$. Refer to section A and B for the remaining details.

### 5.1 EXPAC PREDICTS CLASS-LABELS ACCURATELY

We compare EXPAC with five baseline models in Figure 2. The RAM is a state-of-the-art model for hard attention that classify partially observed images (Mnih et al. (2014)). Our implementation of RAM has a similar structure for the feature extractor $F$, the recurrent net $R$, and the classifier $C$ as EXPAC. Instead of the Partial VAE, RAM has a controller that learns a Gaussian attention policy. We extend RAM by feeding it with a low-resolution gist of an input at the beginning. To ensure that gist is used only to predict glimpse locations and not the class labels, the controller receives input from a separate recurrent net that runs parallel to the recurrent net $R$. We predict the initial hidden state of this additional recurrent net from the gist. We refer to this model as Gist-RAM or GRAM. We also consider a baseline model that attends glimpses on random locations and predicts class labels. Note that EXPAC and the three baselines described so far observe the image only partially

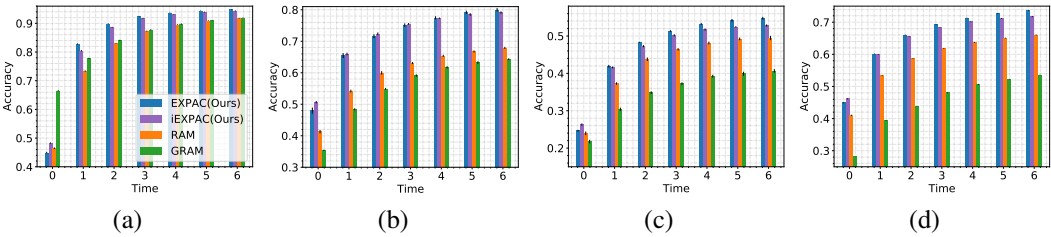

Figure 3: *The classification accuracy as a function of time.* iEXPAC interpolates $EIG$ maps to find optimal glimpses that may overlap. (a) SVHN (b) CIFAR-10 (c) CIFAR-100 (d) CINIC-10.

through a series of glimpses. Additionally, we train a feed-forward CNN and a soft attention model that observe the entire image to predict the class label. For the soft attention model, we adapt the method proposed by Jetley et al. (2018).

The CNN outperforms all other models. The performance of hard attention models surpasses the soft attention model after the former observes a sufficient number of glimpses. Unlike the hard attention models, the soft attention model exhausts its capacity on computing features and attention weights for the uninformative regions. Among the hard attention models, EXPAC shows the best performance. All hard attention models perform better than the random baseline for the digit datasets. For natural image datasets, RAM and GRAM fall behind the random baseline. As observed by Sermanet et al. (2014), RAM does not scale well to natural images, probably due to the unimodal attention policy. Except for the SVHN dataset, GRAM's accuracy is less than RAM's accuracy, despite observing the input gist. We also observed that GRAM overfits the training data and does not generalize well to the test set.

We speculate that regularizing the feature space for classification is very important for the hard attention models. The gradients from the ELBO and the REINFORCE objective regularize this feature space in EXPAC and RAM, respectively. Moreover, the ELBO appears to be a better regularizer than the REINFORCE objective. Zheng et al. (2015) also observed that the use of generative loss regularizes the hard attention model. GRAM uses separate feature spaces for the classifier and the controller; hence the former is not regularized by the latter. GRAM overfits the train set and does not perform well on the test set due to a lack of regularization.

## 5.2 EXPAC PERFORMS WELL WITH THE INTERPOLATED EIG

To let EXPAC find and attend overlapping glimpses with unit stride, we assume a smoothness prior and upsample the $EIG$ map using bicubic interpolation to get EIG values for all overlapping glimpses. We refer to this model as interpolation-EXPAC or iEXPAC. Figure 3 shows a comparison between EXPAC and iEXPAC. iEXPAC performs better than or comparable to EXPAC during initial time-steps. As time progresses, EXPAC observes more input regions than iEXPAC through strided glimpses and outperforms iEXPAC. However, iEXPAC still performs better than RAM and GRAM, which also attends the overlapping glimpses with unit stride. Next, we assess the policies used by these models.

## 5.3 EXPAC LEARNS A USEFUL POLICY FOR CLASSIFICATION

We compared the accuracy of the hard attention models as a function of the area covered in the image. We found that iEXPAC achieves an accuracy similar to the other models by covering less area in the image. The result suggests that iEXPAC observes few but informative regions (refer section D.1). However, this analysis provides only a narrow perspective on the policies used by different models, as each model learns a classifier that adapts to its policy. To better understand the policies, we cover the glimpses observed by the hard attention models and let the baseline CNN (from Figure 2) classify the occluded images. Accuracy should drop if the object of interest is occluded. This analysis is similar to the one performed by Elsayed et al. (2019). The results for natural image datasets are shown in Figure 4 (refer section D.2 for the results on digit datasets). We do not observe any generalizable trend for GRAM. However, we observe that the accuracy drops faster by covering the glimpses attended by the RAM compared to iEXPAC. It appears that the

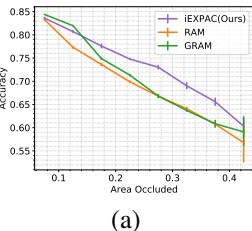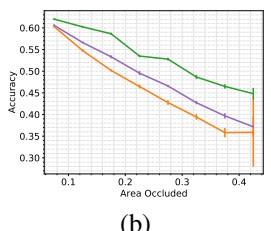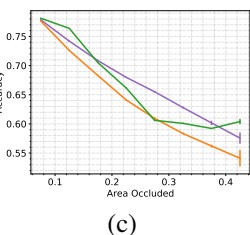

(a)  (b)  (c)

Figure 4: *The classification accuracy of a CNN as a function of the area occluded.* A CNN classifies images with the glimpses attended by various hard attention models occluded. (a) CIFAR-10 (b) CIFAR-100 (c) CINIC-10.

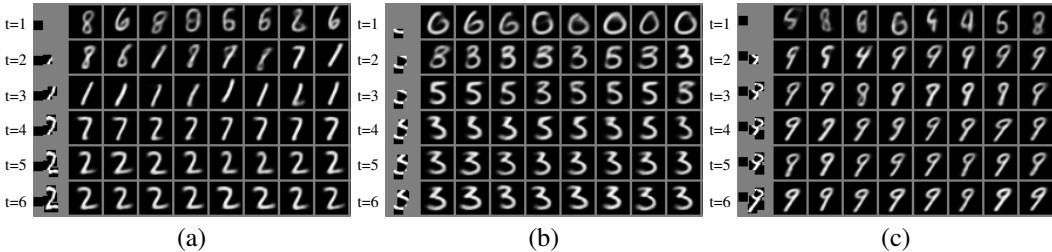

(a)  (b)  (c)

Figure 5: *Visualization of the predictions from the Partial VAE.* In each panel, from left to the right: time step, glimpses observed until time $t$, eight samples of $\tilde{x}$. The content in $\tilde{x}$ on the observed locations is consistent with the glimpses.

RAM attend to the object of interest more frequently than iEXPAC. However, iEXPAC achieves a better performance. The use of EIG in BOED explains the behavior of iEXPAC. If the attended glimpse provides evidence in favor of one class and the Partial VAE also generates features in favor of the same class, then $EIG$ for areas characterizing that class is low. In this case, iEXPAC explores areas with high EIG, away from the regions approving of the believed class. Consequently, iEXPAC often attend regions outside the object of interest. Being a measure of Bayesian surprise, EIG promotes iEXPAC to seek glimpses that seem novel according to the latest belief (Itti & Baldi (2006); Schwartenbeck et al. (2013)). During the process, iEXPAC may collect evidence favoring an actual class that may have initially appeared unlikely and change the decision.

## 5.4 VISUALIZATION OF THE EIG MAPS AND THE ATTENTION POLICY USED BY iEXPAC

We display the generated samples of $\tilde{x}$ for MNIST dataset in Figure 5. For the remaining datasets, we predict $\tilde{f}$. Figure 6 shows the $EIG$ maps and the optimal glimpses found by iEXPAC on CIFAR-10 images. First, notice that the $EIG$ maps are multi-modal. Second, activity in the $EIG$ maps reduces as the model gathers sufficient glimpses to predict a class label. In Figure 6(a), activity in $EIG$ map is reduced as the model settles on a class 'Dog'. Once it discovers left ear of a cat at $t = 3$, the activity in the $EIG$ map increases at $t = 4$. The model changes its decision to 'Cat' once it discovers the right ear at $t = 5$. In Figure 6(b), the model predicts class 'Airplane' at time $t = 3$ once it discovers body of an airplane. At the next time-step, the model expects reduced information gain on the areas where the remaining body of the airplane may lie. In Figure 6(c), the model predicts class 'Deer' once it discovers face of the animal at $t = 4$. Notice reduced activity in the $EIG$ maps at $t = 5$ and 6.

## 5.5 EXPAC ATTAINS HIGHER ACCURACY WITH FLEXIBLE POSTERIOR: AN ABLATION STUDY

We investigate the necessity of a flexible distribution for the posterior $q(z|h_t)$ and, therefore, the necessity of normalizing flows in the encoder $S$. To this end, we model the posterior with a unimodal Gaussian distribution and let $S$ output mean and diagonal covariance of a Gaussian. We do not use flow layers in this case. Figure 7 shows the result for natural image datasets. Refer Figure 11 for results on digit datasets. Modeling a complex posterior using normalizing flows improves accuracy during early time steps. The gap in the accuracy of the two reduces with time. The reason

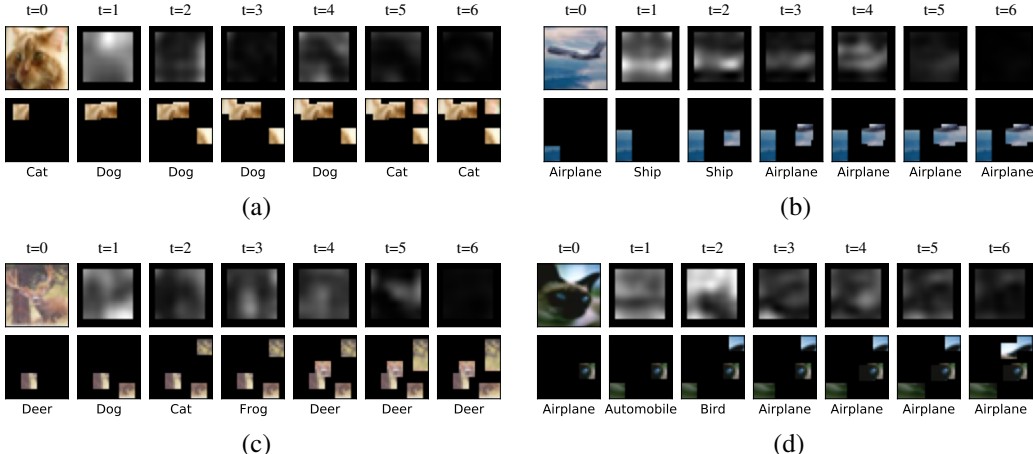

Figure 6: *Visualization of the EIG maps and the glimpses observed by iEXPAC on CIFAR-10 images.* The top row shows the entire image and the $EIG$ maps for time $t = 1$ to 6. The bottom row shows glimpses attended by iEXPAC. The model observes the first glimpse at a random location. We display the entire image for reference; iEXPAC never observes the whole image. (a-c) success cases (d) failure case.

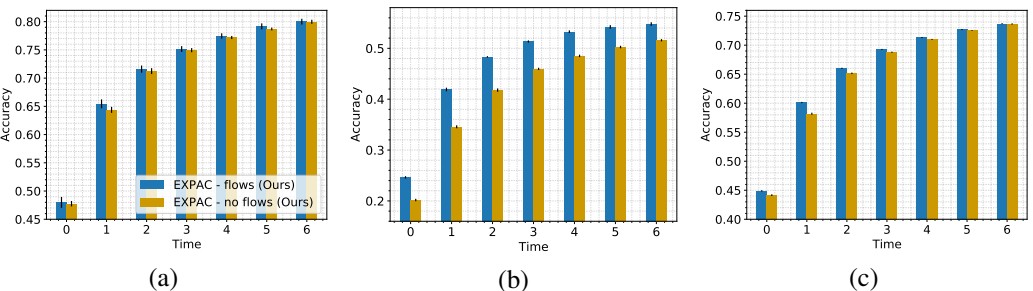

Figure 7: *The classification accuracy as a function of time.* We compare performance of EXPAC with and without using normalizing flows. (a) CIFAR-10 (b) CIFAR-100 (c) CINIC-10.

being as follows. Ideally, the Partial VAE should predict all possibilities of $\tilde{x}$ (or $\tilde{f}$) consistent with the observed region $o_t$. Initially, when EXPAC observes a small region, a complex multi-modal posterior helps determine multiple plausible images from different classes. A unimodal posterior fails to cover all possibilities. As time progress, EXPAC observes a large region, and the associated possibilities decrease. Hence, the performance gap between a complex and a uni-modal posterior decreases. Eventually, only a single possibility of a complete image remains. Then, the performance of a unimodal posterior meets the performance of a complex posterior.

## 6 CONCLUSION

We presented a hard attention model that uses BOED to find the optimal locations to attend when the image is observed only partially. To find an optimal location without observing the corresponding glimpse, the model uses Partial VAE to predict the content of the glimpse. The model predicts the content in either the image space or the feature representation space. The predicted content enables the model to evaluate and compare the expected information gain (EIG) of various candidate locations, from which the model selects a candidate with optimal EIG. The model predicts feature representations and EIG of a subset of glimpses for computational efficiency. We interpolate $EIG$ maps to let the model find optimal glimpses that may overlap. The attention policy used by our model is non-parametric, multimodal, and explainable. We trained our model with a differentiable objective and tested it on five datasets. We found that our model attends glimpses highly relevant to the classification task and achieves better performance than the other baseline models.

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

## A NEURAL ARCHITECTURE

We implement EXPAC for MNIST dataset based on Mnih et al. (2014). We replace $ReLU$ with $LeakyReLU$ and add $BN$ in their architecture. Table 1 shows the neural architecture of EXPAC for color datasets. We implement linear layers using $1 \times 1$ convolution layers. Note that all modules use an indispensable number of layers. $F_l$ is the shallowest network to learn the non-linear representation of the input. $F_g$ uses the least possible layers to achieve the effective receptive field equal to the area of a single glimpse. The decoder $D$ uses the smallest number of $ConvTranspose2d$ and $Conv2d$ to generate the feature maps of required spatial dimensions and refine them based on the global context. We use linear classifiers for SVHN, CIFAR-10 and CINIC-10 datasets and use a two-layer classifier for CIFAR-100. The dimensionality of $f$, $h_t$ and $z$ for various datasets are mentioned in the Table 2.

| | Module | Architecture |
|---|---|---|
| Recurrent feature aggregator | $F_l$ | $Conv2d(k=1) - LeakyReLU - BN - Conv2d(k=1)$ |
| | $F_g$ | $3 \times \{Conv2d(k=3) - LeakyReLU - BN\} - Conv2d(k=2)$ |
| | $F(g_t, l_t)$ | $BN(LeakyReLU(F_g(g_t) + F_l(l_t)))$ |
| | $R(h_{t-1}, f_t)$ | $LN(LeakyReLU(Conv2d(k=1)(h_{t-1}) + Conv2d(k=1)(f_t)))$ |
| Classifier | $C$ | $Conv2d(k=1)$ |
| Partial VAE | $S$ | $4 \times \{ActNorm - permute - NSF\}$ |
| | $D$ | $3 \times \{ConvTranspose2d(k=3) - LeakyReLU - BN\} -$ $4 \times \{Conv2d(k=3, p=1) - LeakyReLU - BN\}$ |

Table 1: Neural architecture of EXPAC for SVHN, CIFAR-10 and CINIC-10 datasets. $k = kernel\_size$, $p = padding$.

| | $f$ | $h_t$ | $z$ |
|---|---|---|---|
| MNIST | 256 | 256 | 256 |
| SVHN | 128 | 512 | 256 |
| CIFAR-10 | 128 | 512 | 256 |
| CIFAR-100 | 256 | 1024 | 256 |
| CINIC-10 | 128 | 512 | 256 |

Table 2: The size of $f$, $h_t$ and $z$.

## B OPTIMIZATION DETAILS

We train the model using Adam optimizer (Kingma & Ba (2014)) with a learning rate of 0.001 and default $\beta_1$ and $\beta_2$, and divide the learning rate by ten on a plateau until convergence. We trained all models on a single Tesla V100 GPU with 16GB of memory for approximately a day. The training procedure for the model shown in Figure 1(b) is shown in Algorithm 2.

## C ALTERNATE GENERATIVE MODEL

We try an alternate generative model for the image $x$ given $h_t$. At time $t$, we divide the image $x$ into two parts, $o_t$ and $u_t$, a set of regions observed by the model and a set of regions that are yet unobserved. The set of observed regions $o_t$ is equivalent to the glimpses observed by the model until time $t$, i.e $o_t = \{g_{1:t}, l_{1:t}\}$. Recall that the hidden state $h_t$ summarizes these glimpses. Hence, we write $p(x|h_t) = p(x|o_t) = p(u_t|o_t)p(o_t)$. For $p(u_t|o_t)$ and $p(o_t)$, we can learn two latent variable models which optimize the ELBO on $\log(p(u_t|o_t))$ and $\log(p(o_t))$. These ELBOs are similar to the ones used to train Neural Processes (Garnelo et al. (2018b;a)) and Partial VAE(Ma et al. (2018)),

---

**Algorithm 2** Training procedure for the model shown in Figure 1(b)

---

1: **while** not converged **do**
2:     $\mathcal{L} = 0$
3:     **for** $t \in \{0, \dots, T-1\}$ **do**                                  ▷ T is the time budget
4:         **if** $t = 0$ **then**
5:             Randomly sample $l_0$
6:         **else** with no gradient:
7:             Compute $\tilde{h}_t$, $p(y|\tilde{h}_t)$ and $EIG$             ▷ Use $\tilde{f}$ and $h_{t-1}$ computed at $t-1$
8:             $l_t = \arg\max\{EIG\}$                              ▷ equation 6
9:         **end if**
10:        Capture $g_t$ at $l_t$; Compute $f_t$, $h_t$, $p(y|h_t)$, $q(z|h_t)$ and $\tilde{f}$
11:        $\mathcal{L} = \mathcal{L} + \mathcal{L}_{\text{PVAE}} + \lambda\mathcal{L}_{\text{CE}}$                       ▷ equation 2,1
12:     **end for**
13:     Compute gradients $\frac{\partial\mathcal{L}}{\partial\theta}$ and update model parameters $\theta$
14: **end while**

---

respectively.

$$\mathcal{L}_{\text{NP}} = \mathbb{E}_{q(\bar{z}|u_t,o_t)}\log(p(u_t|\bar{z},o_t)) - KL[q(\bar{z}|u_t,o_t)||p(\bar{z}|o_t)]$$

$$= \mathbb{E}_{q(\bar{z}|x,h_t)}\log(p(u_t|\bar{z},h_t)) - KL[q(\bar{z}|x,h_t)||p(\bar{z}|h_t)] \tag{8}$$

$$\mathcal{L}_{\text{PVAE}} = \mathbb{E}_{q(z|o_t)}\log(p(o_t|z)) - KL[q(z|o_t)||p(z)]$$

$$= \mathbb{E}_{q(z|h_t)}\log(p(o_t|z)) - KL[q(z|h_t)||p(z)] \tag{9}$$

## C.1 Implementation

We implement a single VAE that takes image $x$ and hidden state $h_t$ as input and reconstructs the image $x$. The design of this VAE is an extension of Partial VAE from Figure 1 with a conditional encoder $E$ and normalizing flows $\bar{S}$. We train this VAE with a combination of ELBOs in equation 8 and equation 9. The prior $p(z)$ is a Gaussian distribution with zero mean and unit variance. A conditional encoder $E$ infers the posterior $q(\bar{z}|x,h_t)$. We use conditional normalizing flows to establish a relation between $q(\bar{z}|h_t)$ and $q(z|h_t)$, i.e. $z = \bar{S}(\bar{z},h_t)$.

$$q(\bar{z}|h_t) = q(z|h_t)\Big|det\Big(\frac{\partial\bar{S}(\bar{z},h_t)}{\partial\bar{z}}\Big)\Big| \tag{10}$$

As in Figure 1, the flow-based encoder $S$ and the decoder $D$ predicts the approximate posterior $q(z|h_t)$ and the complete image $\tilde{x}$, respectively. Given the binary mask $m_t$ as discussed in section 3.1.3, we compute the likelihoods in equation 8 and equation 9 as follows.

$$\log(p(o_t|z)) = -\frac{1}{2}\sum m_t \odot |D(z) - x|^2 \tag{11}$$

$$\log(p(u_t|\bar{z},h_t)) = -\frac{1}{2}\sum(1 - m_t) \odot |D(\bar{S}(\bar{z},h_t)) - x|^2 \tag{12}$$

## C.2 Performance Analysis of the Alternate Generative Model

Here we compare our model's accuracy when the generative model is Partial VAE versus a combination of Partial VAE and Neural Processes. We also train the alternate VAE with just the ELBO of Neural Processes. Figure 8 shows the result. We do not find a single generative model that performs well across datasets. We selected Partial VAE for our model as it does not use an encoder $E$ and $\bar{S}$, which require additional parameters.

# D Additional Results

## D.1 EXPAC attends to glimpses relevant to Classification

We assess the usefulness of the glimpses attended by EXPAC/iEXPAC, RAM, and GRAM for the classification task. Figure 9 shows the models' accuracy as a function of the area observed in an

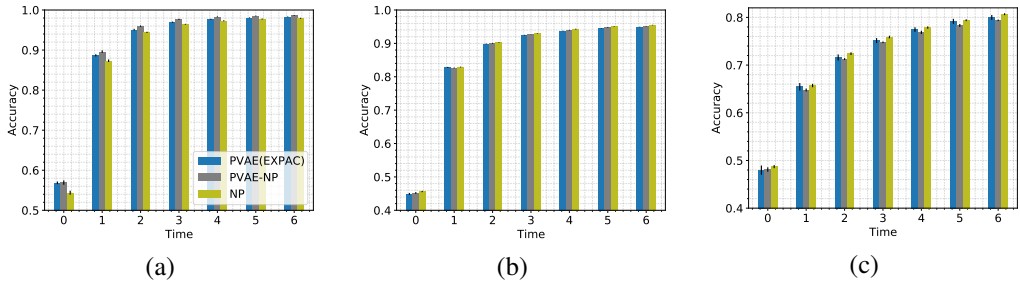

Figure 8: *Classification accuracy as a function of time.* We test various generative models to predict the image content. (a) MNIST (b) SVHN (c) CIFAR-10.

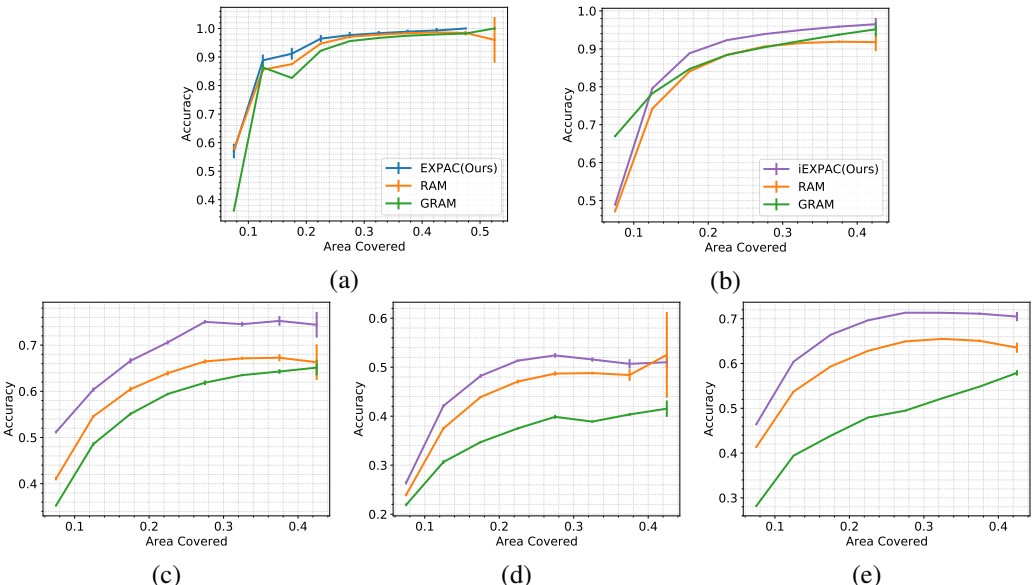

Figure 9: *Classification accuracy of the hard attention models as a function of the area covered.* (a) MNIST (b) SVHN (c) CIFAR-10 (d) CIFAR-100 (e) CINIC-10.

image. Compared to the RAM and GRAM, EXPAC/iEXPAC achieves high accuracy by observing less area. The results indicate that EXPAC/iEXPAC observes regions that are more relevant to the classification task than the other two models. Attending few but task-relevant regions lead to better performance.

## D.2 EXPAC LEARNS A USEFUL POLICY FOR CLASSIFICATION

To get an insight into the attention policies used by different models, we occlude the glimpses attended by the hard attention models and let a baseline CNN predict the class label from an occluded image. We expect a drop in accuracy when the foreground object is occluded. The results for natural image datasets are discussed in section 5.3. Here, we discuss results for the digit datasets. Unlike in the case of natural images, CNN's accuracy for EXPAC/iEXPAC drops at a greater or similar rate as RAM and GRAM. In natural images, iEXPAC explores background regions with high EIG to find evidence for different classes. In digit datasets, the foreground object is at the center and is not related to the surrounding. Hence, EXPAC learns to predict low EIG on the background and always finds an optimal glimpse on the foreground. The accuracy of CNN drops as EXPAC attends to informative glimpses on the foreground. Notice that the accuracy of CNN drops faster for GRAM on the SVHN dataset. The trend suggests that GRAM finds the most informative glimpses. The GRAM also achieves better performance than the RAM for this dataset, as shown in Figure 2.

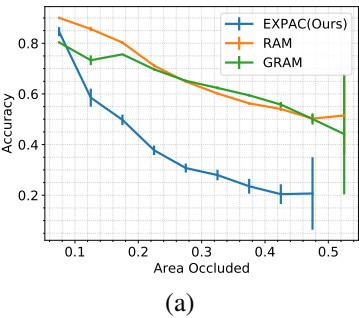 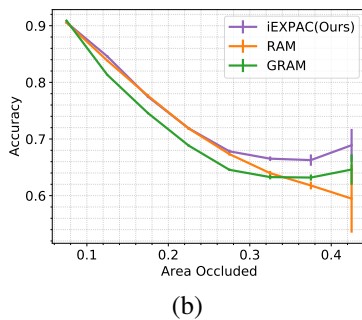

(a)                                                  (b)

Figure 10: *The classification accuracy of a CNN as a function of the area occluded.* A CNN classifies images with the glimpses attended by various hard attention models occluded. (a) MNIST (b) SVHN.

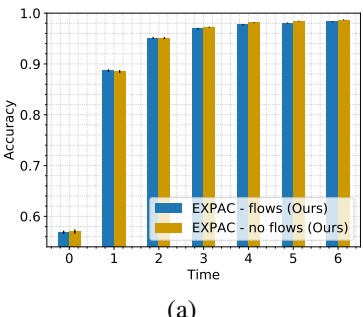 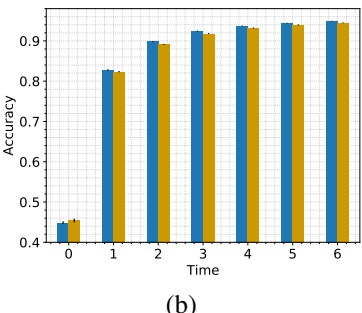

(a)                                                  (b)

Figure 11: *The classification accuracy as a function of time.* We compare performance of EXPAC with and without using normalizing flows. (a) MNIST (b) SVHN.

However, iEXPAC outperforms GRAM. Unlike GRAM, iEXPAC learns a better feature space for classification due to the regularization provided by the generative objective during the training.

