# OpenReview forum: "Achieving Explainability in a Visual Hard Attention Model through Content Prediction"
_ICLR.cc/2021/Conference — Reject_

### Official Review · AnonReviewer1 · 2020-10-27
**paper can be better organized**

**Rating:** 4
**Confidence:** 4

**Review:**

This paper proposes an alternative way to conduct hard attention. Specifically, it estimates the expected information gain (EIG) of attending various regions and at each timestep chooses the region with the maximum EIG. The proposed method is tested on image classification.

Issues:
1. what do you mean "achieving explainability" in the title? I don't quite get it. The visualization in Fig.5 shows only that the region selected in each timestep indeed has the maximum EIG. But how to interpret the explainability from the glimpse sequence is still confusing. I can hardly perceive the sequence using my knowledge.
2. why the proposed method needs three building blocks is not well explained, especially the partial VAE.
3. what is the data used to train the  partial VAE?
4. no ablation study is conducted on these building blocks, such as the normalizing flow module in the partial VAE.
5. I think image classification is not the best task to demonstrate the effectiveness of hard attention method. As shown in the experiments, simply using a standard CNN that takes the whole image as the input obtains the best result.

---

> ### Author Response · Authors · 2020-11-16
> **Response to AnonReviewer1**
>
> We thank the reviewer for devoting their attention to our paper and providing us thorough and thoughtful reviews. We address the reviewer’s queries below.
>
> 1. **Explainability:** The hard attention models make two predictions, the class-label, and the next glimpse location. While the first prediction is explainable by design, the second prediction is not. In this paper, we propose a model that explains the second prediction, i.e., attention policies. Value at location $l$ in the EIG map suggests the amount of information gained in the class probabilities if the model attends to $l$. We can explain that our model seeks a glimpse that seems to provide maximum information about the class-label.
>
> 2. **Building blocks and Partial VAE:** We use a recurrent feature aggregator to extract fixed-length features from a variable number of glimpses, a classifier to predict the class label, and a partial VAE to predict the content yet unobserved. Our model does not have access to the content of the entire image due to the partial observability. The model uses the predicted content to compute the EIG in the absence of actual content.
>
> 3. **Training data for Partial VAE:** We train our entire model in an end-to-end fashion on four image classification datasets: MNIST, SVHN, CIFAR-10, and CINIC-10. Specifically, we use the input image $x$ to evaluate the log-likelihood (eq 3) in the ELBO of the partial VAE.
>
> 4. **Ablation study:** We include an ablation study on the normalizing flows in section 5.5 in the revised manuscript.
>
> 5. **Classification task for hard attention models:** Researchers frequently use classification task to evaluate the visual hard attention models. Many seminal papers in this area demonstrate the effectiveness of the attention models on image classification (Zheng et al. (2015), Larochelle & Hinton (2010), Mnih et al. (2014), Ba et al. (2015), Jaderberg et al. (2015)). **Performance of a CNN:** We expect any efficient model, including CNN, that observes the entire image to attain the best accuracy. We do not expect hard attention models that observe the image partially or at low resolution to match the former's accuracy. The former provides a better perceptive about the comparison of the latter.
>
> We thank the reviewer again and request them to inform us if further clarification is required.
>
> References:
> * Zheng, Yin, et al. "A neural autoregressive approach to attention-based recognition." International Journal of Computer Vision 113.1 (2015): 67-79.
> * Larochelle, Hugo, and Geoffrey E. Hinton. "Learning to combine foveal glimpses with a third-order Boltzmann machine." Advances in neural information processing systems. 2010.
> * Mnih, Volodymyr, Nicolas Heess, and Alex Graves. "Recurrent models of visual attention." Advances in neural information processing systems. 2014.
> * Ba, Jimmy, et al. "Learning wake-sleep recurrent attention models." Advances in Neural Information Processing Systems. 2015.
> * Jaderberg, Max, Karen Simonyan, and Andrew Zisserman. "Spatial transformer networks." Advances in neural information processing systems. 2015.

---

> > ### Comment · AnonReviewer1 · 2020-11-24
> > **Response To Authors**
> >
> > At first thank you for the detailed response to my concerns.
> >
> > My response after reading your respone to my concerns as well as common concerns across reviewers is as follows:
> > 1. Thank you for addressing my concerns regarding the building blocks, the training of partial VAE, evaluation on Classification, and the ablation study on the flow module.
> > 2. As for the explainability, I understand the glimpse sequence is the one that maximizes the EIG. It's better for this paper if you could intuitively explain one image and its resulting glimpse sequence in terms of what kinds of sub-regions are used, or,
> > you may also summarize the common characteristics of sub-regions that are supposed to maximize EIG, e.g. do they all contain objects? I think this intuitions may be useful for researchers from various directions, since EIG is related information gain from an image.
> > 3. While you have emphasized that your focus is achieving *explainability* in the hard attention policies under *partial observability* in the common response, you may improve the paper structure to center around these two key-points, as it looks like all reviewers didn't capture all these two points. For example, CNN may be referred to as completed-observation-based CNN in the experiments, introduction could emphasize the importance of the partial observability setting, etc.
> > 4. Although only Minh's method in 2014  uses partial observations only, you can still come up with more recent baselines to compare with, such as changing the input of those whole-image-based methods to partial observations. Simply claiming they all use whole images as inputs is not very convincing in general.

---

> > > ### Author Response · Authors · 2020-11-25
> > > **Response to AnonReviewer1**
> > >
> > > We thank the reviewer for reaching out to us with additional suggestions. We made improvements in our manuscript following the latest response. We address the reviewer’s comments below.
> > > 1.	We thank the reviewer again for their time and adding value to our paper by recommending an ablation study. We hope that our responses and the revisions addressed the reviewer’s concerns satisfactorily.
> > > 2.	We explain the glimpse sequences for a few images in section 5.4. Note that the images contain a single object; hence one glimpse includes only a small part of an object. We discuss the intuition behind EIG-based glimpse selection at the end of section 5.3. The model finds EIG by anticipating the content of the complete image. At any specific time, the EIG is low for the regions characterizing the believed class and high elsewhere. The model observes glimpses with high EIG in search of new information. As EIG is a measure of Bayesian surprise, it encourages the model to seek the regions that seem novel according to its belief.
> > > 3.	We revised the abstract and the introduction sections to emphasize the partial observability in our problem setting. We also added text in the experiment section to emphasize that the CNN and the soft attention model observes the complete images while other models do not.
> > > 4.	To the best of our knowledge, Saccader is the most recent hard attention model (Elsayed et al. (2019)). The main contribution in the Saccader model is the attention network and the Saccader cell, designed meticulously to aggregate the global context (see equations 1, 2, and 3). Applying Saccader to the partially observable scenes requires significant modifications to its core modules and warrants additional research, which we believe is outside the scope of the presented work. Other methods, such as the ones proposed by Gregor et al. (2015) and Eslami et al. (2016), rely on the Spatial Transformation Network (STN) (Jaderberg et al. (2015)). The STN only functions when complete spatial information is available. Hence, we cannot adopt these methods as well for partially observable scenes.
> > >
> > > Again, we thank the reviewer for their continual feedback to improve our paper. We hope the revised manuscript addresses all concerns of the reviewer.
> > >
> > > References:
> > > * Elsayed, Gamaleldin, Simon Kornblith, and Quoc V. Le. "Saccader: improving accuracy of hard attention models for vision." Advances in Neural Information Processing Systems. 2019.
> > > * Gregor, Karol, et al. "DRAW: A Recurrent Neural Network For Image Generation." ICML. 2015.
> > > * Eslami, SM Ali, et al. "Attend, infer, repeat: Fast scene understanding with generative models." Advances in Neural Information Processing Systems. 2016.
> > > * Jaderberg, Max, Karen Simonyan, and Andrew Zisserman. "Spatial transformer networks." Advances in neural information processing systems. 2015.

---

### Official Review · AnonReviewer3 · 2020-10-28
**Review on "Achieving explainability in a visual hard attention model through content prediction"**

**Rating:** 5
**Confidence:** 4

**Review:**

##########################################################################

Summary:

This paper proposed a new hard attention model for the image classification. They designed hard attention mechanism as a bayesian optimal experimental setting. Compare to other hard attention model, the policies of proposed hard attention can be explainable and differentiable, which is non-parametric. They evaluated their model to four different image classification dataset and their model outperformed than other baseline models.

##########################################################################

Reasons for score:

Even though this is an interesting setting and the technical solutions presented in the paper look reasonable, the idea seems to be pretty incremental as it stacks multiple existing techniques without many innovations.

##########################################################################

Pros:

The proposed hard attention model finds an optimal location using partial variational auto-encoder. Their attention policy is non-parametric and explainable. They validated their model on four different datasets with qualitatively analyzed results.

##########################################################################

Cons:

1. There’s needs for more throughly designed experimental settings with more datasets.
2. Authors need to perform ablation studies with other form of attention mechanisms.

##########################################################################

Questions during rebuttal period:

For me, it's a bit hard to say the proposed methodology is novel.
Authors needs to explain why the proposed model is different from pre-existing methodologies regarding attention mechanism.

#########################################################################

---

> ### Author Response · Authors · 2020-11-16
> **Response to AnonReviewer3**
>
> We thank the reviewer for spending time and providing honest reviews about our paper. And we are glad that the reviewer found the problem setting interesting. We address concerns about the novelty of our work below.
>
> 1. **Datasets:** We evaluate our model on four datasets. The MNIST and SVHN datasets include grayscale and color images of 0-9 digits. The CIFAR-10 and CINIC-10 include color images of natural objects. Furthermore, CINIC-10 includes images from the ImageNet dataset.
> 2. **Ablation Study:** We have added an ablation study on normalizing flows in Section 5.5 in the revised manuscript. All other modules in our model are essential and implemented with an indispensable number of layers, as mentioned in appendix A.
> 3. **Contributions:** Our contributions, especially explainable attention policies for partially observable scenes, set our work apart from the pre-existing methodologies. We note that most existing models observe an entire image and use global information to predict unexplainable attention policies. In contrast, we focus on partially observable scenes where explainable attention policies are critical.  We have added a paragraph at the end of the related works section, which highlights these differences.
> 4. **Novelty:** We agree with the reviewer that the proposed model builds on the existing techniques. However, these techniques are less explored in the context of hard attention. We introduce these techniques to the visual attention community in a principled manner. We emphasize that our paper aims at innovating a unified approach towards hard attention where the attention policies are explainable. In this context, the presented work is unique and novel.
>
> Again, we thank the reviewer and request them to inform us if there are any further questions or suggestions.

---

### Official Review · AnonReviewer2 · 2020-10-28
**An overly complicated system for explainable image classification with hard attentions**

**Rating:** 4
**Confidence:** 1

**Review:**


**Summary:**

This paper follows a less explored strategy for achieving explainability via hard attention. They proposed a recurrent architecture which sequentially observe regions (glimpse) from an image. To decide where to look next, the model maintains a hidden state and use it to estimate the full image (or features of the image). This "content prediction" module allows the model to look ahead and make a decision based on the expected information gain (EIG) over different locations. The objective function (i.e. partial VAE loss and classification loss) in this system is differentiable thus the system can be trained with gradient descent. The authors validated the system on several benchmarks and show comparable performance with baselines.

**Reasons for score:**

The system seems extremely complicated to me, as it involves multiple components and each component by themselves is very complex. However the output of the system is not so appealing either in performance or explainability. Probably I missed something but I don't quite understand the advantages of the proposed system.

**Pros:**

1. The core idea for training the attention policy is intuitive, as the information gain is a natural choice for determining the location of next glimpse. It's also appealing that the predicted content can provide sufficient signals to estimate the information gain.
2. The equation is clear and makes the paper easy to follow.
3. The careful analysis of the experiments is very informative.

**Cons:**

1. In general, my biggest concern about this paper is the complexity of the EXPAC system with the moderate performance. It seems that all benchmark datasets used in the paper are not so difficult (e.g., 10-way classification with 32x32 images) and the performance of the proposed system is still far from satisfactory. Therefore, it's questionable whether the system could be scaled to even more challenging (but more practical) datasets like ImageNet. Also I'm curious about the robustness of training such an intricate system.
2. Regarding explainability, there are quite a few methods (not limited to hard attention) that target the same goal, such as Grad-CAM. However this paper seems to only compare with RAM (and a gist-initialized variation of GRAM). Even if the core algorithm might be different, it's still good to compare with other designs.
(Selvaraju, Ramprasaath R., et al. "Grad-cam: Visual explanations from deep networks via gradient-based localization." Proceedings of the IEEE international conference on computer vision. 2017.)
3. It seems to me that the content prediction module ($S$ and $D$) is critical as the predicted $\tilde{x}$ is used for "lookahead" to estimate the next $l$. However the predicted $\tilde{x}$ is never shown in the paper. I think it would be more straightforward to  better understand the performance of the system by comparing the predicted content with the original.


**Questions during rebuttal period:**

Please address my questions in the cons section.

---

> ### Author Response · Authors · 2020-11-16
> **Response to AnonReviewer2**
>
> We thank the reviewer for assessing our paper and providing invaluable suggestions and comments. We also acknowledge the reviewer for recognizing the spirit of our work i.e. exploring a less explored strategy for explainable hard attention.
>
> ##### Q1
>
> **Complexity:** We note that the complexity of EXPAC is minimal. We use only bare essential modules, namely, a recurrent feature aggregator, a classifier, and a partial VAE. As mentioned in a newly added Appendix A, we implement EXPAC with a small number of layers.
>
> **Robustness:** As mentioned in Appendix B (previously Appendix A), we optimize the model for different datasets with the same hyperparameters, indicating robustness in training our model.
>
> **Performance:** We implement EXPAC with an indispensable number of layers. Adding more layers may help in achieving higher accuracy. We do not invest in performing extensive architecture search. Nonetheless, we endeavor to perform a careful and rigorous analysis of our model.
>
> **Datasets:** Classifying a 32x32 image into ten classes is certainly not difficult when an entire image is observed. However, we observe images through a series of small glimpses, which makes a classification task challenging. Furthermore, we evaluate our model on the CINIC-10 dataset, which includes images from the ImageNet dataset.
>
> **Scalability:** One can implement the classifier and Partial VAE using large-scale models such as ResNet (He et al. (2016)) and NVAE (Vahdat & Kautz (2017)). Note that a classifier and a generator are frequently used for multiple tasks in practice, making them readily available for a hard attention model.
>
> ##### Q2
>
> **Comparison with Grad-CAM:** A feed-forward classifier observes the complete image and predicts the class-label. On the other hand, a recurrent hard attention model actively senses glimpses and makes two predictions, namely, the class-label and the next glimpse location. A seminal work by Selvaraju et al. (2017) explains the first type of model. It cannot explain the prediction of the next glimpse location based on the partially observed scenes as in hard attention. We focus on the latter problem and develop an attention model with in-build explainability. As our problems are fundamentally different, we cannot compare our work with Selvaraju et al. (2017).
>
> ##### Q3
> **Visualization:** We have added Figure 5 in the revised manuscript, displaying the predicted $\tilde{x}$.
>
> We again thank the reviewer for their time and request them to inform us of additional queries.
>
> References:
> * He, Kaiming, et al. "Deep residual learning for image recognition." Proceedings of the IEEE conference on computer vision and pattern recognition. 2016.
> * Vahdat, Arash, and Jan Kautz. "Nvae: A deep hierarchical variational autoencoder." Advances in Neural Information Processing Systems 33 (2020).
> * Selvaraju, Ramprasaath R., et al. "Grad-cam: Visual explanations from deep networks via gradient-based localization." Proceedings of the IEEE international conference on computer vision. 2017.

---

### Official Review · AnonReviewer4 · 2020-10-28
**Review of "Achieving Explainability in a Visual Hard Attention Model through Content Prediction"**

**Rating:** 4
**Confidence:** 3

**Review:**

This paper presents a visual hard-attention image classification model. The difference to standard classification methods such as CNN is that the model provides an explainable inner structure by default, that can be inspected to see what the model focused on. The difference to other state-or-the-art hard-attention models is that this model is differentiable, allowing for more robust and stable optimization.

On a positive note, the presented method is sound and mathematically principled, and the description of it is complete and technically correct. The paper is also well written, well organized, and easy to read. The relevant related work is cited.

However, the paper suffers from two major flaws.
Firstly, the contribution of the proposed method with respect to other recent hard-attention models based on reinforcement learning it is not well motivated - other than that this model is differentiable. The last paragraph in the Related Work provide no statement whatsoever as to what the present method contributes over the latest methods in the literature.
Secondly, the baseline hard-attention model in the experiments, (Mnih et al. 2014), is very old and it is not surprising that the proposed method outperforms it. A more interesting baseline would be a later hard-attention model such as (Elsayed et al. 2019). Moreover, the used datasets are all quite simplistic, and it would be more interesting with a more realistic one.

Due to the above, the recommendation is Reject - but the authors are strongly encouraged to do experiments on more challenging data and compare to a newer baseline.

---

> ### Author Response · Authors · 2020-11-16
> **Response to AnonReviewer4**
>
> We are grateful to the reviewer for considering our work and providing us invaluable feedback. We also thank the reviewer for recognizing and acknowledging the strengths of our paper. We address the reviewer's concerns below.
>
> **Contribution:** We delineate our contributions in the last paragraph of the introduction section. Most recent hard attention models based on reinforcement learning observe entire images and use unexplainable attention policies. In contrast, we present *explainable attention policies* for *partially observable* scenes, which is a valuable contribution. As suggested by the reviewer, we have added a paragraph at the end of the related works section, highlighting our contribution to the latest methods.
>
> **Comparison:** Saccader, a great model presented by Elsayed et al. (2019), observes an entire image and gathers global contextual information in the attention network. In contrast, we observe an image only partially through a series of glimpses. As our model does not use global contextual information, we cannot make a fair comparison between the two methods. We compare our method with Minh et al. (2014) as they also work with partially observable scenes.
>
> **Datasets:** Note that we focus on developing a systematic approach to explainable attention policies under partial observability. The partial observability in our problem setting imposes additional constraints, making classification task on simplistic datasets more challenging. Furthermore, the CIFAR-10 and CINIC-10 are real-world image datasets. The latter also includes images from the ImageNet dataset.
>
> We thank the reviewer for their time and request them to inform us of further questions.
>
> References:
> * Elsayed, Gamaleldin, Simon Kornblith, and Quoc V. Le. "Saccader: improving accuracy of hard attention models for vision." Advances in Neural Information Processing Systems. 2019.
> * Mnih, Volodymyr, Nicolas Heess, and Alex Graves. "Recurrent models of visual attention." Advances in neural information processing systems. 2014.

---

### Author Response · Authors · 2020-11-16
**Common Response**

We thank the reviewers for providing thoughtful comments and suggestions. Here we highlight our main contribution, namely achieving *explainability* in the hard attention policies under *partial observability*. We believe this contribution will prove extremely valuable to the visual attention community.

For a task of image classification, the hard attention models make two predictions that need explanation. First, the models predict the class label based on the observed glimpses, which is explainable by design (Elsayed et al. (2019)). Second, they predict the next attention-worthy glimpse location, for which they do not provide any explanation. We recognize the need to address this important problem based on the observation noted at the end of the first paragraph in the introduction section. Unlike the existing models, we can explain why our model chose to attend a specific location. We use the EIG maps, as displayed in Figure 6 (previously Figure 5), to provide this explanation. The model chooses to attend the glimpses that seem to maximize the EIG in the class label. *The EIG being maximum at the location $l_t$ should be interpreted as a rationale behind selecting a glimpse at $l_t$.*

We would also like to highlight the partial observability in our problem setting. Like Minh et al. (2014), we observe an image only partially through a series of glimpses to predict the class-label and the attention-worthy location. In contrast, recent works observe an entire image at high resolution and use global context to find attention-worthy locations (Elsayed et al. (2019), Xu et al. (2015), Jaderberg et al. (2015)). Unlike these models, the models like ours and Minh et al. (2014) enjoy a broader range of applications. For instance, consider a mobile edge device that senses the scene through a series of partial observations captured using a moving camera. Here, we can employ our model to guide the camera's movement in the direction of the important region and recognize the scene using partial observations.

Lastly, we note that our aim in this paper is to innovate a principled approach towards explainable hard attention under partial observability. We make a unique contribution by presenting a unified framework for this task. We evaluate our model on four datasets and show promising results. We expect our work to kindle interest and stimulate additional research around explainable hard attention.

We again thank the reviewers for their time and address their queries separately.

References:
* Elsayed, Gamaleldin, Simon Kornblith, and Quoc V. Le. "Saccader: improving accuracy of hard attention models for vision." Advances in Neural Information Processing Systems. 2019.
* Xu, Kelvin, et al. "Show, attend and tell: Neural image caption generation with visual attention." International conference on machine learning. 2015.
* Mnih, Volodymyr, Nicolas Heess, and Alex Graves. "Recurrent models of visual attention." Advances in neural information processing systems. 2014.
* Jaderberg, Max, Karen Simonyan, and Andrew Zisserman. "Spatial transformer networks." Advances in neural information processing systems. 2015.

---

> ### Author Response · Authors · 2020-11-25
> **Additional results on CIFAR100 dataset**
>
> Following the common concern by the reviewers, we include results on a more challenging dataset, CIFAR-100. The dataset contains images from one hundred categories. Identifying a correct class-label based on a small number of glimpses is extremely difficult. Nonetheless, our model consistently outperforms the baseline methods on this dataset. The results suggest the robustness and generalizability of our model.
>
> The presented attention model achieves significantly better results on CIFAR-10 and CIFAR-100 datasets compared to the attention-less CNN-based classification methods when faced with incomplete information or occlusion (Zhong et al. (2020), Osherov et al.(2017)). These papers report error rates of around 30% and 60% for CIFAR-10 and CIFAR-100 under the occlusion level of 0.2. Our model achieves approximately 20% and 45% error rates for these datasets under the effective occlusion level greater than 0.5 (See Figure 2(c) and (d)). The low error rates achieved by our model proves the effectiveness of the proposed hard attention mechanism.
>
> References:
> * Osherov, Elad, and Michael Lindenbaum. "Increasing cnn robustness to occlusions by reducing filter support." Proceedings of the IEEE International Conference on Computer Vision. 2017.
> * Zhong, Zhun and Zheng, Liang and Kang, Guoliang and Li, Shaozi and Yang, Yi. “Random Erasing Data Augmentation.” Proceedings of the AAAI Conference on Artificial Intelligence (AAAI). 2020.

---

### Author Response · Authors · 2020-11-25
**Final Revision**

Once again, we thank the reviewers for devoting their attention to our paper. We revised the manuscript based on the reviewers' feedback. Here we describe the new elements of the paper.

1.	We included results on the CIFAR-100 dataset. This dataset is more challenging compared to the other datasets as it contains one hundred classes. The consistent performance on a more challenging task proves the robustness of the proposed method.
2.	We revised the abstract and the introduction section to emphasize the partial observability in our problem setting.
3.	We added a paragraph at the end of the related works section to highlight our contributions over the recent methods in the literature.
4.	We included an ablation study on the normalizing flows in section 5.5.
5.	We added Figure 5, displaying the predictions by the Partial VAE.
6.	We included details about the architecture of our model in Appendix A.

---

### Decision · Program_Chairs · 2021-01-07
**Final Decision**

**Decision:**

Reject

**Comment:**

This paper proposes a new hard attention model for the image classification as a way to achieve explainability. Two of the reviewers do not find the output of the system interpretable, which is a fatal weakness for a paper on XAI.
R1: The visualization in Fig.5 shows only that the region selected in each timestep indeed has the maximum EIG. But how to interpret the explainability from the glimpse sequence is still confusing. I can hardly perceive the sequence using my knowledge.
R2: However the output of the system is not so appealing either in performance or explainability.
R3: Post-discussion note: For me, it's a bit hard to say the proposed methodology is novel. Authors needs to explain why the proposed model is different from pre-existing methodologies regarding attention mechanism.
R4: Due to the above, the recommendation is Reject - but the authors are strongly encouraged to do experiments on more challenging data and compare to a newer baseline.